# A Coherent Integrated TDOA Estimation Method for Target and Reference Signals

**Xinxin Ouyang** [1,2,*], **Shanfeng Yao** [2] **and Qun Wan** [1]

1 School of Information and Communication Engineering, University of Electronic Science and Technology of China, Chengdu 611731, China
2 National Key Laboratory of Science and Technology on Blind Signal Processing, Chengdu 610041, China
* Correspondence: ouyxxwork@163.com

**Abstract:** The performance of a time difference of arrival (TDOA) localization system is severely affected by time synchronization errors, and making use of reference signals is a common solution for the problem. The traditional method has two steps, first to measure the TDOAs of the target signal and reference signal separately, and next, to compensate the estimated target TDOA with the difference of the estimated reference TDOA and the true reference TDOA. Since the performance of the TDOA estimation is mainly decided by the frequency information, a coherent integration TDOA estimation method for the target signal and reference signal is proposed in this paper, based on cross correlation phase difference compensation, with use of the signals' frequencies. First, as per the traditional method, the separated cross correlation functions of the target signal and reference signal were obtained by cross correlation, and the target TDOA and reference TDOA of the separate method were estimated. Next, the cross correlation phase was analyzed for each signal. Then the coherent integration cross correlation was obtained with phase compensation, from which the estimation of the target TDOA and reference TDOA could simultaneously be achieved. We performed simulation comparisons with the two methods, and showed that the proposed algorithm provided better performance.

**Keywords:** TDOA estimation; passive localization; time synchronization error; integration estimation; reference station





## 1. Introduction

The passive localization of radio emitters has many advantages in wireless communication, sonar, radio astronomy, radar and seismology, and has been an important aspect of tracking and location research in recent years [1–17]. Passive localization of signal emitters can be achieved by various location systems; hyperbolic location is one of the important types of passive location systems, and the subject of much research. Hyperbolic location is also called TDOA location, locating an emitter with TDOA measurements between three or more stations. Most of the research has focused on TDOA estimation and localization algorithms [1–16], some attention has also been paid to hardware, such as TDOA sensors [17]. The process of TDOA location usually includes two steps, first to measure the TDOA parameters, depending on the emitter location of the signal, and then to use them to estimate the emitter location.

It is obvious that TDOA localization performance depends on the accuracy of TDOA parameters' estimation. Additionally, the performance of TDOA localization is also tightly coupled with the types of system errors, such as time synchronization error, clock error and sensor position error [18–23].

Assuming that the clock error and sensor position error have been corrected, this paper will focus on the correction of the time synchronization error. A time synchronization error can be introduced by hardware or software latency. It is one of the challenges faced by TDOA localization, since time synchronization errors can directly affect the accuracy of the

TDOA estimation. Precise time synchronization for localization systems can be established by specially dedicated hardware, such as GPS receivers and atomic clocks, which may be very expensive. Therefore, many papers have paid attention to passive localization without time synchronization [24–28]. Anthony and Weiss made use of arbitrary signals to establish synchronization between stations [24,25]. Xu proposed a time synchronization method for a TDOA localization system with two-signal sensing and sample counting [26]. Assuming that the time and frequency offsets were constant throughout the measurement period, Yeredor used consecutive TDOA and frequency difference of arrival (FDOA) measurements to jointly estimate the transmitter location and the offsets parameters [27]. Park suggested a round-trip time-based method that could estimate the internal delay of node with another time-reference node [28]. Methods in [24–26] can be called non-cooperative reference signal or station, and [29,30] also researched the cooperative signal or broadcasts to achieve better time synchronization.

The accuracy of TDOA measurements depends mainly on the frequency distribution of signals, and some research has been conducted on multi-carrier signals and frequency-hopping signals which made use of the wide band of such signals to improve performance [31–34]. The existing TDOA localization systems synced by reference signals processes the target signal and reference signal separately. These methods cannot make full use of the frequency information of the target signal and reference signal, and the TDOA estimation performance for the target signal is not optimal.

For the purpose of high accuracy TDOA localization, we discuss, in this paper, the coherent integration of the TDOA estimation method for target and reference signals. Using the received signals model, the cross correlation function of the target signal and reference signal were derived, and the phase relation was analyzed. With cross correlation phase difference compensation, coherent integration of the target signal and reference signal cross correlation was obtained, and the performance improvements of the TDOA estimation for target signal were proven by simulations. The proposed method can be applied to TDOA localization systems synced by reference signals.

The paper is organized as follows. Section 2 introduces the received signal model for TDOA localization systems synced by reference signals, the cross correlations of target signal and reference signal are analyzed separately, and the phases of the cross correlations are revealed. In Section 3, the traditional separate method for target TDOA estimation is first provided, then the coherent integration method through phase difference compensation is derived. In Section 4, the performance is demonstrated via several numerical simulations. Discussion and conclusions are provided in Sections 5 and 6.

## 2. Signal Model and Cross Correlation Analysis

We assume that target signal and reference signal are received simultaneously at two spatially separated receivers in the presence of noise. The complex envelopes of the monitored signals at the two receivers can be modeled as:

$$\begin{cases} r_1(t) = \gamma_1 e^{j\varphi_1}[s_T(t) + s_R(t)] + n_1(t) \\ r_2(t) = \gamma_2 e^{j\varphi_2}[s_T(t - \tau_T - \Delta) + s_R(t - \tau_R - \Delta)] + n_2(t) \end{cases} \tag{1}$$

where $s_T(t)$ and $s_R(t)$ represent the complex envelopes of the target signal and reference signal waveforms, $\gamma_1$ and $\gamma_2$ are the attenuation factors at the two receivers, $\varphi_1$ and $\varphi_2$ are the random phases produced by the mixers of the two receivers [35], $\tau_T$ and $\tau_R$ are the TDOA of the target signal and reference signal between the two receivers, $\Delta$ is the time synchronization error between the two receivers, and $n_1(t)$ and $n_2(t)$ are white Gaussian noise which are uncorrelated with $s_T(t)$ and $s_R(t)$.

Assuming that the carrier frequencies of the target signal and the reference signal are $f_T$ and $f_R$, and their bandwidths are $B_T$ and $B_R$, respectively, then the received signal model can be converted to:

$$\begin{cases} X_1(f) = \gamma_1 e^{j\varphi_1}[S_T(f) + S_R(f)] + N_1(f) \\ X_2(f) = \gamma_2 e^{j\varphi_2}[S_T(f)e^{-j2\pi f(\tau_T+\Delta)} + S_R(f)e^{-j2\pi f(\tau_R+\Delta)}] + N_2(f) \end{cases} \tag{2}$$

Since the target signal and the reference signal occupy different frequency bands, they can be separated through suitable filters. Their cross correlation function can be obtained by separated cross correlation equations. Assuming that the signals have flat spectrums, the target signal cross correlation can be described as:

$$\begin{aligned} R_T(\Delta\tau) &= \gamma_1\gamma_2^* e^{j(\varphi_1-\varphi_2)} \int_{f_T-\frac{B_T}{2}}^{f_T+\frac{B_T}{2}} S_T(f)S_T^*(f)e^{-j2\pi f(\tau_T+\Delta)}e^{j2\pi f\Delta\tau}df + W_T \\ &= \gamma_1\gamma_2^* e^{j(\varphi_1-\varphi_2)} \int_{f_T-\frac{B_T}{2}}^{f_T+\frac{B_T}{2}} \left|S(f)\right|^2 e^{j2\pi f(\Delta\tau-\tau_T-\Delta)}df + W_T \\ &= \gamma_1\gamma_2^* e^{j(\varphi_1-\varphi_2)} E_T \text{sinc}(B_T(\Delta\tau-\tau_T-\Delta))e^{j2\pi f_T(\Delta\tau-\tau_T-\Delta)} + W_T \end{aligned} \tag{3}$$

Then the cross correlation of the reference signal is:

$$R_R(\Delta\tau) = \gamma_1\gamma_2^* e^{j(\varphi_1-\varphi_2)} E_R \text{sinc}(B_R(\Delta\tau-\tau_R-\Delta))e^{j2\pi f_R(\Delta\tau-\tau_R-\Delta)} + W_R \tag{4}$$

where $E_T$ and $E_R$ are the energy of the target signal and reference signal, respectively, and $W_T$ and $W_R$ are produced by noise. Ignoring the influence of noise, peak values will be obtained at $\Delta\tau = \tau_T + \Delta$ and $\Delta\tau = \tau_R + \Delta$, respectively, and the phases of $R_T(\Delta\tau)$ and $R_R(\Delta\tau)$, here, are zero.

In actual application, the target signal and reference signal usually need to be converted to baseband signal with digital down conversion (DDC). After converting to baseband signal, the received target signal model will be changed to:

$$\begin{cases} x_{1,T}(\text{t}) = \gamma_1 e^{j\varphi_1} s_T(t) \cdot e^{-j2\pi f_T t} \\ x_{2,T}(\text{t}) = \gamma_2 e^{j\varphi2} s_T(t-\tau_T-\Delta) \cdot e^{-j2\pi f_T t} \end{cases} \tag{5}$$

Then the cross correlation of the target signal will be transformed into:

$$\begin{aligned} R_T(\Delta\tau) &= \gamma_1\gamma_2^* e^{j(\varphi_1-\varphi_2)} \int_{-\frac{B_T}{2}}^{\frac{B_T}{2}} S_T(f)S_T^*(f)e^{-j2\pi(f+f_T)(\tau_T+\Delta)}e^{j2\pi f\Delta\tau}df + W_T \\ &= \gamma_1\gamma_2^* e^{j(\varphi_1-\varphi_2)} \int_{-\frac{B_T}{2}}^{\frac{B_T}{2}} \left|S(f)\right|^2 e^{j2\pi f(\Delta\tau-\tau_T-\Delta)}e^{-j2\pi f_T(\tau_T+\Delta)}df + W_T \\ &= \gamma_1\gamma_2^* e^{j(\varphi_1-\varphi_2)} E_T \text{sinc}(B_T(\Delta\tau-\tau_T-\Delta))e^{-j2\pi f_T(\tau_T+\Delta)} + W_T \end{aligned} \tag{6}$$

When the frequency center is converted to zero, the cross correlation of the target signal removes the item $e^{j2\pi f_T\Delta\tau}$, and the phase of $R_T(\Delta\tau)$ will no longer be zero.

After the same process, the cross correlation of reference signal will be transformed into:

$$R_R(\Delta\tau) = \gamma_1\gamma_2^* e^{j(\varphi_1-\varphi_2)} E_R \text{sinc}(B_R(\Delta\tau-\tau_R-\Delta))e^{-j2\pi f_R(\tau_R+\Delta)} + W_R \tag{7}$$

## 3. Integrated TDOA Estimation Method

### 3.1. Traditional Method

When the TDOA localization system performance is severely affected by time synchronization errors, reference signals are used for correction. The steps of the traditional method are as follows.

First, the TDOAs of the target signal and reference signal are separately estimated, with cross correlation:

$$\begin{cases} \Delta\hat{\tau}_T = \underset{\Delta\tau}{\text{argmax}}\left|R_T(\Delta\tau)\right| \\ \\ \Delta\hat{\tau}_R = \underset{\Delta\tau}{\text{argmax}}\left|R_R(\Delta\tau)\right| \end{cases} \tag{8}$$

Next, the estimated target TDOA will be corrected with the difference between the estimated reference TDOA and the true reference TDOA. Since the reference signal emitter position is known, the true reference TDOA $\tau_R$ is known. Therefore, the difference between the estimated reference TDOA and the true reference TDOA is equal to the estimated time synchronization error:

$$\hat{\Delta} = \Delta\hat{\tau}_R - \tau_R \tag{9}$$

Therefore, the estimated target TDOA would be:

$$\hat{\tau}_T = \Delta\hat{\tau}_T - \hat{\Delta} \tag{10}$$

### 3.2. Coherent Integration Method

TDOA estimation performance mainly depends on the frequency distribution, target signal and reference signal occupying different frequency bands, therefore, TDOA estimation performance may be improved with an efficient coherent integration process.

In the following analysis, it can be seen that random phases introduced by mixers for the target signal and the reference signal are the same, and the different parts of the cross correlation phases of the target signal and reference signal are produced by different carrier frequencies and TDOAs. At the true TDOA value, one is $-2\pi f_T(\tau_T + \Delta)$, and the other is $-2\pi f_R(\tau_R + \Delta)$. To obtain coherent integration of the two cross correlations, the phase difference must be compensated to ensure that the phases of the two cross correlations at the true value $\Delta\tau_T = \tau_T + \Delta$ and $\Delta\tau_R = \tau_R + \Delta$ are the same.

The fact is that the true $\Delta\tau_T$ and $\Delta\tau_R$ are unknown, so the cross correlation phases of the target signal and reference signal are unknown. However, different $\Delta\tau_T$ and $\Delta\tau_R$ can be searched to compensate the different phases of the two cross correlations by multiplying $e^{j2\pi f_T \Delta\tau_T}$ and $e^{j2\pi f_R \Delta\tau_R}$, respectively. When the searched $\Delta\tau_T$ and $\Delta\tau_R$ are equal to the true values of $\Delta\tau_T = \tau_T + \Delta$ and $\Delta\tau_R = \tau_R + \Delta$, the phases of the two cross correlations are zero, so coherent integration can be achieved.

Then the coherent integration cross correlation can be described as:

$$
\begin{aligned}
R_{CI}(\Delta\tau_T, \Delta\tau_R) &= R_T(\Delta\tau_T)e^{j2\pi f_T \Delta\tau_T} + R_R(\Delta\tau_R)e^{j2\pi f_R \Delta\tau_R} \\
&= \gamma_1 \gamma_2^* e^{j(\varphi_1 - \varphi_2)} E_T \operatorname{sinc}(B_T(\Delta\tau_T - \tau_T - \Delta))e^{-j2\pi f_T(\tau_T + \Delta)}e^{j2\pi f_T \Delta\tau_T} + W_T \\
&\quad + \gamma_1 \gamma_2^* e^{j(\varphi_1 - \varphi_2)} E_R \operatorname{sinc}(B_R(\Delta\tau_R - \tau_R - \Delta))e^{-j2\pi f_R(\tau_R + \Delta)}e^{j2\pi f_R \Delta\tau_R} + W_R \\
&= \gamma_1 \gamma_2^* e^{j(\varphi_1 - \varphi_2)} \big[ E_T \operatorname{sinc}(B_T(\Delta\tau_T - \tau_T - \Delta))e^{j2\pi f_T(\Delta\tau_T - \tau_T - \Delta)} \\
&\quad + E_R \operatorname{sinc}(B_R(\Delta\tau_R - \tau_R - \Delta))e^{j2\pi f_R(\Delta\tau_R - \tau_R - \Delta)} \big] + W_T + W_R
\end{aligned} \tag{11}
$$

Therefore, the coherent integration TDOA estimations with time synchronization error is:

$$(\Delta\hat{\tau}_T, \Delta\hat{\tau}_R) = \underset{\Delta\tau_T, \Delta\tau_R}{\operatorname{argmax}} \left| R_{CI}(\Delta\tau_T, \Delta\tau_R) \right| \tag{12}$$

From Equation (11), we can see that when:

$$
\begin{cases}
\Delta\tau_T = n/f_T + \tau_T + \Delta, & n = 0, \pm 1, \pm 2, \cdots \\
\Delta\tau_R = n/f_R + \tau_R + \Delta, & n = 0, \pm 1, \pm 2, \cdots
\end{cases} \tag{13}
$$

the coherent integration $R_{CI}(\Delta\tau_T, \Delta\tau_R)$ has period peaks for $\Delta\tau_T$ and $\Delta\tau_R$. The period peaks will severely affect the TDOA estimation performance when the signal-to-noise ratio (SNR) is low.

We can also see that when:

$$f_T(\Delta\tau_T - \tau_T - \Delta) = f_R(\Delta\tau_R - \tau_R - \Delta) \tag{14}$$

$$\Delta\tau_T = f_R/f_T \cdot \Delta\tau_R + \tau_T + \Delta - f_R/f_T \cdot (\tau_R + \Delta) \tag{15}$$

the coherent integration $R_{CI}(\Delta\tau_T, \Delta\tau_R)$ will reach a peak ridge. The peak ridges will also degrade the performance of TDOA estimation.

The coherent integration time synchronization error and target TDOA estimations can be obtained from Equation (12).

$$\begin{cases} \hat{\Delta} = \Delta\hat{\tau}_R - \tau_R \\ \hat{\tau}_T = \Delta\hat{\tau}_T - \hat{\Delta} \end{cases} \tag{16}$$

If the phase difference has not been compensated, the non-coherent integration cross correlation can be obtained as:

$$R_{NCI}(\Delta\tau_T, \Delta\tau_R) = |R_T(\Delta\tau_T)| + |R_R(\Delta\tau_R)| \tag{17}$$

Then the non-coherent integration time synchronization error and target TDOA estimation can be achieved as per the coherent integration steps.

### 4. Numerical Results

In order to examine the performance of the proposed coherent integration TDOA estimation method, denoted by *CI*, Monte Carlo simulations were performed in this section, and the results were compared to the separated estimation method, the non-coherent integration method, and the Cramer–Rao bounds (CRB) for separated estimation method, which was calculated according to the results in [2]. The simulation parameters are listed in Table 1.

**Table 1.** Simulation parameters.

| Parameters | Value |
|---|---|
| Prior Sample Rate | 1 MHz |
| Sample Rate after DDC | 100 kHz |
| Reference Signal Bandwidth | 50 kHz |
| Target Signal Bandwidth | 20 kHz |
| Sample Time | 10 ms |
| Target Signal Carrier Frequency | 400 kHz |
| Reference Signal Carrier Frequency | 100 kHz |

During the simulation, the SNRs of the target signal and reference signal were varied from 10 dB to 20 dB. For each SNR, 500 Monte Carlo experiments were performed. The simulation results of the separated estimation method and non-coherent integration method were denoted by SE and NCI, respectively. The simulation results are shown in Figures 1–4.

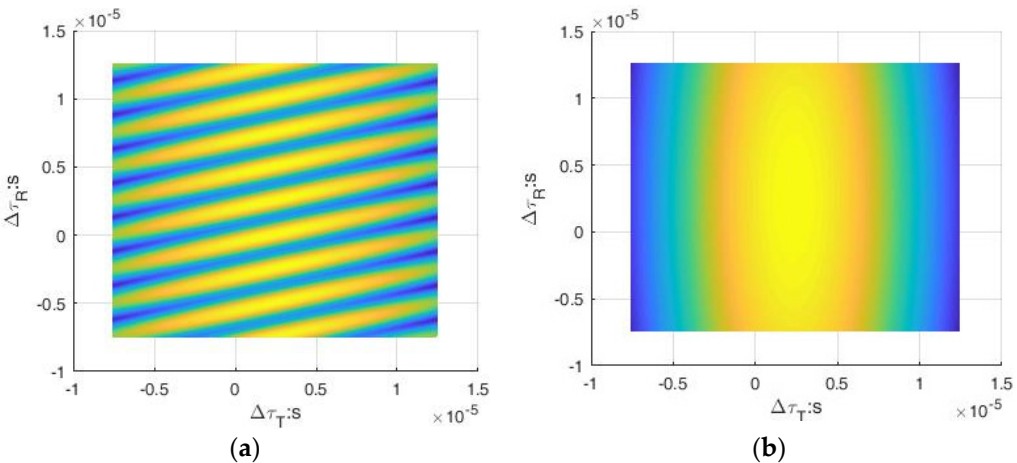

**Figure 1.** Integrated cross correlation spectrum at SNR = 20 dB. (**a**) Coherent integration. (**b**) Non-coherent integration.

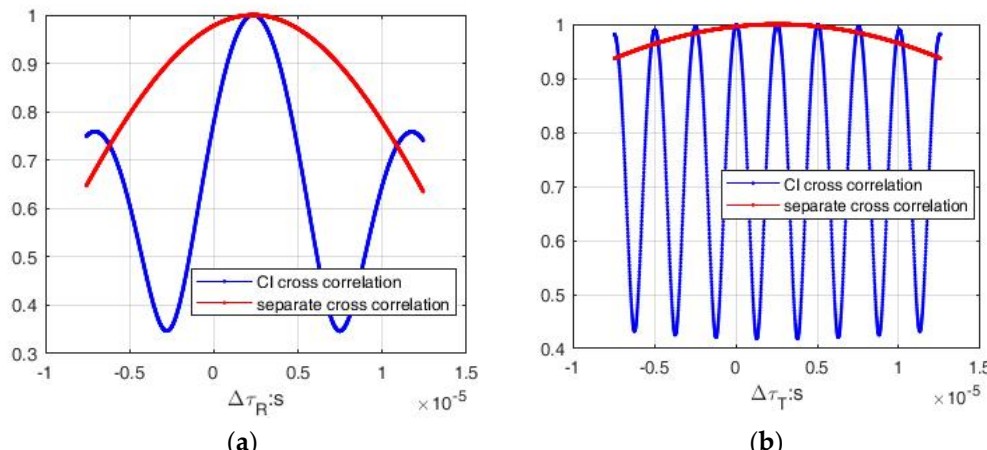

**Figure 2.** Normalized cross correlation comparison of coherent integration and separate process at SNR = 20 dB. (**a**) Reference signal cross correlation. (**b**) Target signal cross correlation.

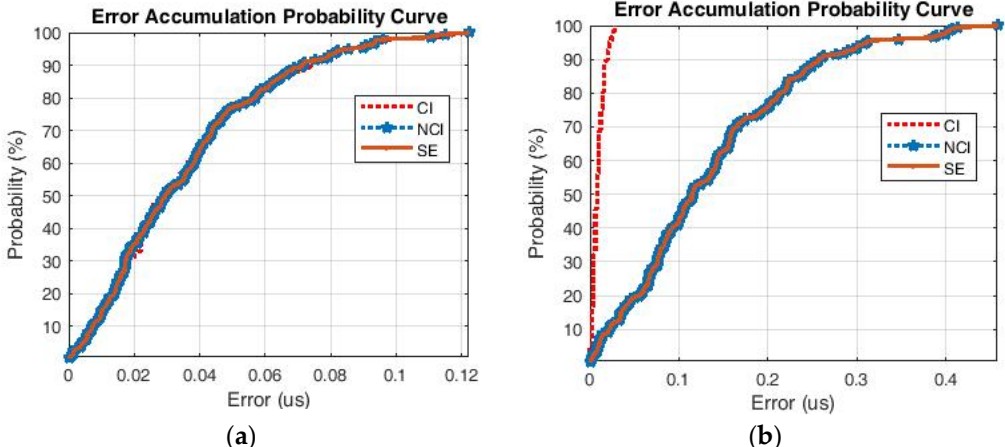

**Figure 3.** Estimation error accumulation probability curves of the three methods at SNR = 20 dB. (**a**) System time synchronization error estimation. (**b**) Target TDOA estimation.

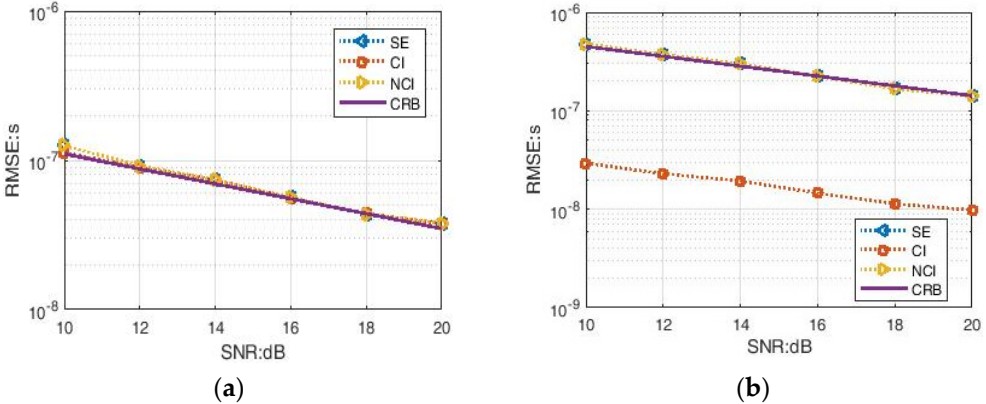

**Figure 4.** Estimation performance comparison of the three methods and CRB. (**a**) System time synchronization error estimation. (**b**) Target TDOA estimation.

Figure 1 shows the comparison of the coherent and non-coherent integration cross correlation spectrum, $R_{CI}(\Delta\tau_T, \Delta\tau_R)$ and $R_{NCI}(\Delta\tau_T, \Delta\tau_R)$ at SNR = 20 dB. We can see that the coherent integration spectrum has a sharper main lobe than the non-coherent integration spectrum, which may bring better estimation performance. However, the

coherent integration spectrum has many period peak ridges, as analyzed in Section 3, while the non-coherent integration spectrum does not.

Figure 2 shows the normalized cross correlation comparison of coherent integration and separate process. The normalized cross correlations of coherent integration, in Figure 2, are the slices of $R_{CI}(\Delta\tau_T, \Delta\tau_R)$ as shown in Figure 1, at the true value $\Delta\tau_T = \tau_T + \Delta$ and $\Delta\tau_R = \tau_R + \Delta$. The slices of $R_{NCI}(\Delta\tau_T, \Delta\tau_R)$ at the true value $\Delta\tau_T = \tau_T + \Delta$ and $\Delta\tau_R = \tau_R + \Delta$ are the same when calculated with the results of the separate method, so they have not been shown in Figure 2. From the simulation results, we can see that the coherent integration cross correlations of target signal and reference signal have period peaks, but the main lobe is sharper than the separate cross correlation.

The estimation errors distribution accumulation probability curves of the three methods—coherent integration, non-coherent integration and separate method—at SNR = 20 dB are shown in Figure 3. We can see that the error distribution of non-coherent integration and the separate method are nearly the same, both for time synchronization error estimation and target TDOA estimation. The estimation error distribution of time synchronization error for coherent integration is almost the same as the other two methods. The maximum estimation error of time synchronization error for *CI* is 118 ns, and for the other two methods is 122 ns. In the case of target TDOA estimation, the coherent integration method performs significantly better than the other two methods. The maximum estimation error of target TDOA for *CI* is 28 ns, but is larger than 450 ns for the two other methods.

Figure 4 shows the comparison of CRB and the estimated root mean square error (RMSE) results of the three methods. The CRBs of the TDOA estimation for single signals have been well researched, such as in [1,2,34,35]. According to [2], the CRB of the TDOA for a signal with a rectangular spectrum is:

$$\sigma_{DTO} = \frac{1}{\beta}\frac{1}{\sqrt{BT\gamma}} \approx \frac{0.55}{B}\frac{1}{\sqrt{BT\gamma}} \qquad (18)$$

where $B$ is the noise bandwidth at the receiver input, assumed to be the same for both receivers, $\beta$ is the "rms radian frequency" in the received signal spectrum, $T$ is the signal duration, and $\gamma$ is the effective input SNR, as defined by:

$$\frac{1}{\gamma} = \frac{1}{2}\left[\frac{1}{\gamma_1} + \frac{1}{\gamma_2} + \frac{1}{\gamma_1\gamma_2}\right] \qquad (19)$$

where $\gamma_1$ and $\gamma_2$ are the SNRs in the respective receivers. Then we can obtain the CRBs for the reference signal and the target signal, denoted by $\sigma_{DTO,R}$ and $\sigma_{DTO,T}$. Concerning the separated estimation method, the time synchronization error estimation only uses the reference signal, as shown in Equations (8) and (9), so the CRB of time synchronization error estimation is the same as the reference signal, that is, $\sigma_{DTO,R}$, while the target TDOA estimation must use both the reference signal and the target signal, as shown in Equations (8)–(10). Since the target signal is uncorrelated to the reference signal, the CRB of the target TDOA estimation should be $\sqrt{\sigma^2_{DTO,R} + \sigma^2_{DTO,T}}$.

The CRBs in Figure 4 were calculated for the SE method of a single signal, making independent use of the parameters of the target signal or reference signal. While the *CI* method uses the information of the two signals based on coherent integration, its theoretical performances and CRBs need further research, therefore, the CRBs for the *CI* method are absent in this manuscript. Since the *CI* method uses the frequency information of two signals, the occupied frequency band is wider than the single target signal, and the main lobe is much sharper than the separated estimation method, as shown in Figure 2. Therefore, the RMSE of *CI* is much lower than the result of SE, while the CRB is calculated for SE, thus, the RMSE of *CI* is much lower than the CRB in Figure 4b.

From Figure 4, it can be seen that, for the time synchronization error estimation, the coherent integration method performed nearly the same as the other two methods.



However, for the target TDOA estimation, the coherent integration method performed better than the others, with performance improved by more than 10 times compared with the separate method.

## 5. Discussion

A coherent integration and direct estimation method for target TDOA and time synchronization error has been proposed. The work was based on the hypothesis that the random phases introduced by the mixer and receiver were the same for signals at different frequency bands received simultaneously by the receiver. The simulation results showed that the performance improvement of target TDOA estimation was significantly better than the time synchronization error estimation, but the reason for this still needs more research. Furthermore, only CRBs for SE were provided here; clarification is still needed as to how the theoretical performances and CRB are affected by signals' parameters, for the *CI* method.

## 6. Conclusions

Accurate TDOA estimation is the basis of passive TDOA localization systems. Focusing on high accuracy to target the TDOA estimation problem for localization systems with time synchronization errors, this paper proposed a coherent integration cross correlation method for the target signal and reference signal, to improve the performance of target TDOA estimation. The target TDOA estimation performance of the proposed algorithm significantly outperformed that of the traditional separate method. The case of non-coherent integration method was also analyzed, and its performance was found to be the same as the traditional separate method.

**Author Contributions:** Conceptualization, X.O. and Q.W.; methodology, X.O.; software, X.O. and S.Y.; validation, X.O. and S.Y.; formal analysis, X.O.; investigation, X.O.; data curation, X.O. and S.Y.; writing—original draft preparation, X.O.; writing—review and editing, S.Y.; visualization, X.O. and S.Y.; supervision, Q.W. All authors have read and agreed to the published version of the manuscript.

**Funding:** This research received no external funding.

**Conflicts of Interest:** The authors declare no conflict of interest.

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
