# Peer review of "A Coherent Integrated TDOA Estimation Method for Target and Reference Signals"

_electronics, doi:10.3390/electronics11162632_

Round 1

Reviewer 1 Report

Please find the attacched PDF file.

Reviewer 2 Report

In this paper, authors propose a coherent integration method to estimate and compensate the timing error for the target signal in uplink TDoA.

The major concerns and questions about this paper are as follows:

1. In second row of Eq.(3), the coefficients  in front of the integral is missing.

2. Same in Eq. (3), from second row to third row, |S(f)|^2 is a function of f and shall not be taken out from the integral directly. The derivation from second row to third row is satisfied only when "|S(f)|^2 is a constant throughout the frequency band" which does not make sense in practice. Please correct me if my statement is wrong.

3. Based on the above concern, Eq. (4), Eq. (6), Eq. (7) and Eq. (11) would be disqualified.

4. In Figure 4, the RMSE of different methods are compared with Cramer-Rao lower bound (CRB or CRLB). How did you obtained the CRB? Why CI performance in Figure 4(b) stays below CRB? CRB is the bound of minimum error you can achieve. How did you obtain the result that the RMSE of CI is much lower than CRB?

Reviewer 3 Report

Dear Authors,

The article presents the CI method for the TDOA estimation and compares the results with non-CI methods. The topic is interesting to the readers however it seems that the authors lost the effort for a discussion and conclusion. 

In the scientific texts is suggested to avoid "obviously," "sometimes," "much," "we know," etc ... 

In the text are acronyms without explanation.

line 83 - receiver outputs?

line 195 needs an explanation

I suggest after each figure follows the text with an explanation

Figures - caption: according to the title, explain to the reader why is each figure important

line 212-218 need to rephrase

discussion and conclusion are weak 

I advise rephrasing lines 223-224.

Brgds.

Round 2

Reviewer 2 Report

Major concerns have been addressed, I have no further questions.

Please merge your response 4 in cover letter into manuscript for a better clarification of the Figure 4(b).

Author Response

Thanks very much for your valuable comments, the explanation after Figure 4 in the revised manuscript has been merged with the response 4 in cover letter .